# Functional Identification of Olfactory Receptors of *Cnaphalocrocis medinalis* (Lepidoptera: Crambidae) for Plant Odor

**DOI:** 10.3390/insects14120930

**Published:** 2023-12-07

**Authors:** Jianjun Cheng, Jiawei Gui, Xiaoming Yao, Hong Zhao, Yujie Zhou, Yongjun Du

**Affiliations:** 1Institute of Pesticide and Environmental Toxicology, Zhejiang University, 866 Yuhangtang Road, Hangzhou 310058, China22116229@zju.edu.cn (J.G.); 2Zhejiang Plant Protection, Quarantine and Pesticide Management Station, Hangzhou 310029, China; xmyao@126.com; 3Agricultural Technology Extension Center of Shengzhou, Shengzhou 312400, China; 4Agricultural Technology Extension Center of Zhuji, Zhuji 311800, China; zhouyujie1@126.com

**Keywords:** functional identification, *Cnaphalocrocis medinalis*, floral odorant, olfactory receptor

## Abstract

**Simple Summary:**

The rice leaf roller, *Cnaphalocrocis medinalis,* is a migratory insect pest threatening rice production. Monitoring its population dynamics is a critical component of the integrated management system for this pest. Plant odor-trapping has been used as a tool for population monitoring. However, there is a lack of knowledge on the molecular mechanism by which the *C. medinalis* moth recognizes the plant odor. In the present study, we amplified 12 odor receptor genes and tested differential expression of these genes in male and female moths of *C. medinalis*. We identified the function of three olfactory receptor genes in *C. medinalis.* CmedOR31 was a specific receptor for geraniol, and CmedOR32 was a broad-spectrum OR receptor that responds to both foliar odor and floral scent. The ligands of CmedOR1 were linalool, phenethyl alcohol, 2-phenylethanol, and phenylacetaldehyde. CmedOR1 belonged to a unique conserved OR lineage that strongly responded to phenylacetaldehyde in the evolution of *Glossata* species, and the OR1 expression in *C. medinalis* antennae was significantly higher in females than in males. CmedOR1 was the odor receptor for *C. medinalis* adults to locate food sources.

**Abstract:**

*Cnaphalocrocis medinalis* (Lepidoptera: Crambidae) is a migratory insect pest on rice crops. The migratory *C. medinalis* population in a particular location may be immigrants, local populations, emigrants, or a mix of these. Immigrants are strongly attracted to plant odor. We conducted research to identify the olfactory receptors in a floral scent mixture that is strongly attractive to *C. medinalis*. Through gene cloning, 12 olfactory receptor (OR) genes were amplified and expressed in Xenopus oocytes in vitro, and three of them were found to be responsive to plant foliar and floral volatiles. These were CmedOR31, a specific receptor for geraniol; CmedOR32, a broad-spectrum OR gene that responded to both foliar and floral odors; and CmedOR1, which strongly responded to 10^−4^ M phenylacetaldehyde. The electrophysiological response to phenylacetaldehyde was extremely high, with a current of 3200 ± 86 nA and an extremely high sensitivity. We compared the phylogenetic tree and sequence similarity of CmedOR genes and found that CmedOR1 belonged to a uniquely conserved OR pedigree in the evolution of *Glossata* species, and the ORs of this pedigree strongly responded to phenylacetaldehyde. The expression of OR1 was significantly higher in the females than in the males. Localization of CmedOR1 in the antennae of *C. medinalis* by fluorescence in situ hybridization showed that CmedOR1 was expressed in both males and females. CmedOR1 may be an odor receptor used by females to locate food sources. The function of these ORs and their role in pest monitoring were discussed.

## 1. Introduction

The rice leaf roller, *Cnaphalocrocis medinalis* (Lepidoptera: Crambidae), is an important migratory pest of rice that decreases photosynthesis through leaf rolling, resulting in a reduction in rice yield [1]. In China, a technique called disturbing and moth-counting in the early morning is commonly used for population monitoring, but it is difficult to make an accurate count and the process is labor-intensive [2]. Alternatively, other methods, such as pheromone trapping and floral odor trapping, are used for pest monitoring [3]. Monitoring *C. medinalis* by sex pheromone trapping [4,5] is species-specific. The sex pheromone trapping in rice fields is related to the physiological status of *C. medinalis*, whereas the physiological status is strongly influenced by migration status, seasonality, and ovarian and testis development [6]. These factors can add complexity to the sex pheromone trapping method and reduce the stability of the monitoring results. The odor of host plant leaves or flowers can attract both male and female pests [7,8,9,10,11,12]. Migratory *C. medinalis* adults usually need to have supplemental nutrition after their emergence [13]. Thus, the attractiveness of floral scents is worth studying, but studies on these odors have focused on their attraction to some Lepidoptera and Diptera pests [14,15]. However, unlike the sex pheromone, the scent of flowers is not species-specific because it also attracts non-target pests, pollinators, and some natural enemy species [16,17]. Therefore, floral trapping has predominantly been used for biodiversity surveys [18].

Olfactory sensilla on the insect antennae detect odor molecules in the environment and then generate electrical signals that are transmitted to the central nervous system for processing to determine the final behavior [19]. Odorant molecules are diffused through pores located on the surface of the sensilla, where odorant-binding proteins (OBP) carry them through the water-soluble lymph to the odorant receptors (ORs) in the dendrites of olfactory receptor neurons (ORN). Then, the activation of OR by odor molecules provides a series of electrical signals, which are further transmitted to the central nervous system [20,21,22]. ORs play a vital role in the olfactory system of insects. In *C. medinalis*, a large number of olfactory receptor repertoires have been identified through transcriptome and gene expression analysis [23,24,25,26]. The function of some chemosensory proteins was characterized by RNA interference [27,28,29]. Functional analysis of ORs in some lepidoptera using a Xenopus oocyte expression system coupled with a two-electrode voltage clamp revealed that these ORs are either pheromone receptors (PRs) in male antennae tuned to sex pheromone components or general ORs in both male and female antennae responding to plant foliar volatiles and floral scents [30,31,32,33]. 

Previous research indicated that *C. medinalis* adults were strongly attracted to plant odor [3,34]. The objective of the present study was to identify and characterize the general ORs in *C. medinalis* according to the plant odor. 

## 2. Materials and Methods

### 2.1. Insect Collection

*Cnaphalocrocis medinalis* adults used in this study were captured with a sweeping net in a rice field (30°18′12.93″ N, 120°34′24.75″ E) in Shaoxin, China. The adult antennae, head, thorax, abdomen, legs, labipalpi, and genitalia were separated under a sterile condition, frozen immediately in liquid nitrogen, and stored at −80 °C until use.

### 2.2. RNA Isolation and RACE Amplification

Total RNA was extracted from each of the *C. medinalis* body parts in the Trizol Reagent (Invitrogen, Waltham, MA, USA). The RNA extraction was carried out following the manufacturer’s instructions. Single-stranded cDNA templates were synthesized with 1 μg of total RNA using a SuperScript™ III First-Strand Synthesis System (Thermo Fisher Scientific, Waltham, MA, USA). Based on the partial sequence previously identified by analyzing the transcriptome data of *C. medinalis* antennae [24,35], primers were designed to amplify the core sequence by RT-PCR. Based on the core sequences, a 5′/3′ RACE Kit, 2nd Generation (Roche, Santa Clara, CA, USA) was used to obtain the full-length sequence of the *C. medinalis* OR (CmedOR) genes. All primers used in this article are showed in the Appendix A.

### 2.3. Vector Construction and cRNA Synthesis

Specific primers were designed with the Kozak consensus sequence and homologous sequence of pT7Ts cut by the restriction enzyme EcoRV to amplify the full open reading frames (ORFs) of the 12 CmedORs (CmedOR1, CmedOR2, CmedOR26, CmedOR27, CmedOR28, CmedOR31, CmedOR32, CmedOR33, CmedOR39, CmedOR40, CmedOR41, and CmedOR46). The amplified ORFs were cloned into the expression vector pT7Ts using a Takara Infusion Cloning Kit (Takara, Japan) based on the manufacturer’s instructions. The extracted plasmids were linearized by digestion with EcoRI and used as templates for cRNA synthesis using the mMESSAGE mMACHINE T7 Kit (Ambion, Austin, TX, USA). The purified cRNAs were diluted with nuclease-free water to a concentration of 2 µg/µL and stored at −80 °C.

### 2.4. Quantitative PCR (qPCR)

The relative expression levels of CmedOR transcripts in male and female antennae were determined using qPCR on a CFX connect real-time system (Bio-Rad, Hercules, CA, USA). Each qPCR volume (25 μL) contained 12.5 μL of TB Green Premix Ex Taq II (Takara, Japan), 0.5 μL of each primer (10 μM), 1 μL of cDNA, and 10.5 µL of Easy Dilution (Takara, Japan). The qPCR cycling conditions were an initial denaturation at 95 °C for 30 s, followed by 39 cycles of 95 °C for 5 s and 60 °C for 30 s. Actin (GenBank JN029806.1) and tubulin genes [26] were used as internal controls to verify the integrity of the cDNA templates. This experiment was repeated three times using three independent RNA samples. Gene expression levels relative to the mean of actin and tubulin were calculated using the 2^−ΔΔCT^ method. ΔCT = CT_OR gene_ − the mean of CT_actin_ and CT_tubulin_ gene, ΔΔCT = ΔCT_different tissues_ − ΔCT_minimum._

### 2.5. Electrophysiological Recording

The electrophysiological recording was carried out as described by Cao et al. [30]. Mature healthy Xenopus oocytes (stages V–VII) were selected and then treated with 1.5 mg/mL LiberaseTM Research Grade (Roche, USA) in washing buffer (96 mM NaCl, 2 mM KCl, 5 Mm MgCl_2_, and 5 Mm HEPES) for 15 min at 25 °C. A mixture of CmedORco and CmedOR genes (total 27.6 ng) was microinjected into the oocytes. After injection, oocytes were cultured for 4–6 days at 18 °C in 1×Ringer’s buffer (96 mM NaCl, 2 mM KCl, 5 mM MgCl_2_, 0.8 mM CaCl_2_, and 5 mM HEPES [pH 7.5]) supplemented with 5% dialyzed horse serum, 50 mg/mL tetracycline, 100 mg/mL streptomycin, and 550 mg/mL sodium pyruvate. Chemical compounds were dissolved in dimethyl sulfoxide at a concentration of 1 mol/L, then diluted at different concentrations in 1×Ringer. The whole-cell currents were recorded with a two-electrode voltage clamp and an OC-725C oocyte clamp (Warner Instruments, Hamden, CT, USA) at a holding potential of −80 mV. Two electrodes are both glass electrodes with a tip diameter of 3 to 7 μm. The glass electrode was filled with 3 mol/L KCl solution, and the electrode resistance was 0.1~1 MΩ. The reference electrode was connected to the cell pool by using an Ag/AgCl electrode. The sampling frequency was 2000 Hz. After each of the plant odor compounds (Table 1) diluted in 1×Ringer was flown to stimulate the oocytes for 15 s, the current response was recorded, and 1×Ringer was then used to wash the oocytes until the current response was back to baseline. Data acquisition and analysis were carried out with Digidata 1550B and pCLAMP 10.0 (Axon Instruments Inc., Union City, CA, USA).

*Xenopus laevis* endogenous receptors and channels require the existence of a follicular membrane. In our experiments, we performed pre-treatment on *Xenopus laevis* oocytes, such as collagenolysis to eliminate the follicular membrane, so when we were conducting compound testing, all current responses of the blank group were 0.

### 2.6. Fluorescence In Situ Hybridization

Localization of CmedOR1, CmedOR31, and CmedOR32 in the antennae of *C. medinalis* was undertaken by triple fluorescence in situ hybridization. To visualize the expression pattern of these CmedORs in *C. medinalis* antennae, their RNA antisense probes were detected by anti-digoxigenin-peroxidase in combination with FITC-Tyramide, CY5-Tyramide, and iF647-Tyramide, respectively. Their RNA antisense probes were synthesized by Yuanmu bio-tech company (Shanghai, China). *C. medinalis* antennae were dissected and collected in 4% paraformaldehyde in phosphate-buffered saline (PBS). After a fixation at 4 °C for 24 h, the antennae were dehydrated, embedded in paraffin, and sliced into 4 μm sections at −60 °C with a cryostat microtome (LEICA, CM1850) and then air-dried at 62 °C for 20 min. The dried slices were boiled in the retrieval solution for 10–15 min and then naturally cooled. Proteinase K (20 ug/mL) working solution was added to cover the objectives for incubation at 37 °C for 15 min. After incubation, the slices were washed with pure water and then washed three times with PBS (pH 7.4) in a Rocker device for 5 min each. Endogenous peroxidase was inactivated by a 15 min incubation in 3% methanol-H_2_O_2_. A pre-hybridization solution to each section was added to incubate at 37 °C for 1 h. After the pre-hybridization solution was removed, the probe hybridization solution (1 µM) was added and incubated in a humidity chamber for hybridization at 42 °C overnight. Slides were then washed in 2× SSC at 37 °C for 5 min, 1× SSC twice for 5 min each at 37 °C, and 0.5× SSC at 25 °C for 10 min, respectively. After the washing, slides were blocked by incubation for 50 min with anti-digoxigenin-peroxidase (Jackson, MS, USA). Slides were then washed three times in PBS buffer. Tyramide amplification was carried out according to the instructions of the TSA kit (Servicebio, Wuhan, China). The slides were then visualized under a laser-scanning microscope (NIKON ECLIPSE CI). Pictures were adjusted in terms of contrast and brightness using Photoshop (Adobe systems, San Jose, CA, USA).

### 2.7. Statistical Analysis

SPSS 16.0.2 (SPSS Inc. 2008, Chicago, IL, USA) was used to analyze the data in this study. The resulting data regarding expression levels of the CmedOR gene and electrophysiological recordings were analyzed using a one-way ANOVA. Pairs of treatment means in the expression levels of the CmedOR gene were compared and separated by the Student’s *t*-test (*p* < 0.05), whereas pairs of treatment means in electrical recordings were compared and separated by the Duncan’s multiple range test (*p* < 0.05).

## 3. Results

### 3.1. Quantitative Expression of C. medinalis OR Genes

Twelve CmedOR genes were, respectively, tested in the seven different body parts of C. medinalis female and male adults, and their quantitative expression levels are shown in Figure 1. All of the CmedORs were detected in the antennae. Of these, five genes (CmedOR1, CmedOR27, CmedOR28, CmedOR30, and CmedOR33) were strongly expressed in the females compared to the males. The greatest sex-specific difference in antennal gene expression was in CmedOR28, in which the expression level was about 23-fold greater in females than in males. In contrast, CmedOR2, CmedOR26, CmedOR41, and CmedOR46 were strongly expressed in the male antennae. The expression level of CmedOR41 was up to 6.4-fold greater in males than in females. However, the expression levels of CmedOR31, CmedOR32, and CmedOR40 were similar in both the female and male antennae.

A phylogenetic tree was constructed by CmedOR genes and other lepidopteran insect ORs (Figure 2). We also analyzed HarmOR42 and HassOR42, which evolved from a non-ditrysian species at the base of the Glossata lineage, and for which the orthologs retained a possible conserved phenylacetaldehyde-sensing function across the Glossata species. CsupOR2 was also in this highly conserved lineage, and CmedOR1 was derived from the same branch, suggesting that CmedOR1 may also be an ortholog to these genes. We did a homology analysis of amino acid sequences on CmedOR1, HarmOR42, HassOR42, and CsupOR2. Results showed that these genes shared a great sequence identity (84.10%) (Figure 3), and this supported our hypothesis.

### 3.2. Functional Characterization of CmedOR Genes in the Xenopus Oocyte Expression System

The 12 ORs were co-expressed with CmedORco in Xenopus oocytes, and their responses to 26 plant odor compounds (Table 1) are shown in Figure 4. Among the genes tested, 3 ORs had detectable responses to these compounds (0.1 μM). Compared to specific ORs, broad-tune ORs only generated currents of 20–50 nA. Notably, CmedOR1 had the strongest response to phenylacetaldehyde (3200 ± 86 nA) (Figure 4a,b). Dosage-response studies showed that even low concentrations (10^−7^ M) of phenylacetaldehyde elicited significant responses (89 ± 6 nA) (Figure 4c,d). CmedOR1 also responded to linalool, a common volatile compound emitted from host flowers and leaves, with the current at 64 ± 5 nA (0.1 μM), and two isomers, phenethyl alcohol and 2-phenethyl alcohol, with a current at 130 ± 10 nA (Figure 4a,b). CmedOR31 responded specifically to geraniol (Figure 4e,f), with the current up to 128 ± 16 nA, whereas CmedOR32 showed nonspecific responses to plant foliar odorants cis-3-hexenyl acetate, E2-hexenal, and β-caryophyllene, as well as floral volatile compounds methyl 2-methoxybenzoate, β-myrcene, and ethyl butyrate (Figure 4g,h).

### 3.3. Fluorescence In Situ Hybridization

Results of fluorescence in situ hybridization for the location of CmedOR1, CmedOR31, and CmedOR32 in *C. medinalis* antennae are shown in Figure 5. To visualize the expression pattern of CmedOR1, CmedOR31, and CmedOR32 in *C. medinalis* antennae, RNA antisense probes were detected by anti-digoxigenin-peroxidase in combination with FITC-Tyramide, CY5-Tyramide, and iF647-Tyramide, respectively (Figure 5). CmedOR1, an olfactory receptor used for locating food sources, was widely expressed in both male and female antennae (green signal). Consistent with the qPCR results, females expressed more CmedOR1 than males. According to a previous study on the sensilla of *C. medinalis* [36], CmedOR1 was shown in the trichoid sensilla both in males and females. Compared with CmedOR1, CmedOR31 (red signal) and CmedOR32 (pink signal) were expressed at much lower levels.

## 4. Discussion

We used three criteria to filter OR genes in this study: high expression levels, long fragments, and homology analysis of amino acid sequences with other insects’ ORs whose functions have been identified. However, the general odor receptors for detecting plant volatiles have low homology, which is different from sex pheromone receptor genes in conservation. Except for CmedOR1, which we found and is a unique conserved general odor receptor gene, the rest of the ORs in *C. medinalis* moths do not have much in common in the homology of genes, so in order to study the odor recognition mechanism in *C. medinalis* for plant volatiles, it is necessary to test different OR genes. Overall, we selected these 12 OR genes in the present study.

In the present study, a total of 26 plant volatile compounds, some of which were formulated to make a floral scent mixture with strong attractiveness to *C. medinalis* [34], were tested on 12 *C. medinalis* OR genes amplified through gene cloning and expression in vitro. Three ORs responded to floral odors (Figure 4). CmedOR31 was a specific receptor for geraniol (Figure 4e,f), and CmedOR32, a broad-spectrum OR, responded to both plant foliar volatiles and floral scents (Figure 4g,h). The response of CmedOR1 to 10^−4^ M phenylacetaldehyde was greater than 3200 ± 86 nA (Figure 4b), indicating extremely high sensitivity. We compared the phylogenetic tree and sequence similarity of CmedOR genes and found that CmedOR1 belonged to the uniquely conserved OR pedigree in the evolution of *Glossata* species (Figure 2 and Figure 3), and the ORs of this pedigree had a strong response to phenylacetaldehyde. Results from the expression of OR genes revealed that CmedOR1 was much more common in *C. medinalis* females than in males (Figure 1), suggesting that CmedOR1 may be an odor receptor used by females to locate food sources for supplemental nutrition. Localization of CmedOR1 in the antennae of *C. medinalis* by fluorescence in situ hybridization showed that CmedOR1 was expressed in the long trichoid sensilla of males but in the short trichoid sensilla of females (Figure 5). In addition, CmedOR31 and CmedOR32 were only observed in the female antennae (Figure 5). These results would help us understand the olfactory response of *C. medinalis* to the volatiles emitted from host leaves and flowers.

The volatile compounds of flowers, such as phenylacetaldehyde, phenethyl alcohol, and methyl salicylate, have been reported to attract some Lepidoptera adults [14,15,37,38,39,40,41]. The odors also attracted many pollinating insects [42,43,44], natural enemies [16], Hymenoptera, Hemiptera [44], and Coleoptera species [17]. Phenylacetaldehyde is a common component of floral scent in many angiosperms [45,46] and an honest floral signal to pollinators for the quality of nectar in *Brassica rapa* [42,47]. As an alternative to sex pheromone trapping, floral trapping has potential in pest management [48,49,50]. In a two-year monitoring study using floral trapping, the attractiveness to *C. medinalis* was greater or similar to the level of sex pheromone trapping [3,34]. As a migratory pest, *C. medinalis* often needs supplemental nutrition from food sources such as nectar [13]. Therefore, understanding the molecular mechanism of olfactory recognition in *C. medinalis* for floral odors, especially the function of olfactory receptors, would be helpful to improve the pest monitoring method. The synthetic blend from plant volatiles could be further optimized and improved. Floral odor attracts both female and male moths, while the sex pheromone can only attract males. However, our previous research has demonstrated that for *C. medinalis,* female moths in the immigratory and local generations, mostly trapped by the floral odor, were those mated and egg-laid, whereas the females releasing sex pheromones to attract males were those in their calling period [34]. Therefore, the physiological and migratory status of adult *C. medinalis* moths in the field could be differentiated by two types of attractants, namely the floral odor and the sex pheromone.

Li et al. [26] presented transcriptome data for the olfactory-related genes of *C. medinalis*. ORco genes [51] and OR genes of *C. medinalis* were subsequently reported [24]. However, in the identification of the olfactory gene function of *C*. *medinalis*, most experiments have been focused on odorant-binding proteins (OBPs) [27] and chemosensory proteins (CSPs) [52]. In the present study, 12 complete ORFs of OR genes were amplified, and three OR gene ligands were found in components of host plant foliar and floral odors. Among them, CmedOR1 was found to be extremely sensitive to phenylacetaldehyde, and it also responded to linalool (Figure 4a–d). Previous studies also found that these two compounds were attractive to the migratory velvet bean caterpillar moth (*Anticarsia gemmatalis*) [14]. Phylogenetic tree and amino acid sequence analyses revealed that CmedOR1 belonged to the highly conserved OR pedigree in the evolution of *Glossata* species (Figure 2 and Figure 3). Its similarity to HarmOR42, HassOR42, and CsupOR2 was as high as 84.10%, and the OR genes in this pedigree also had a strong response to phenylacetaldehyde [46]. Geraniol is a ligand of CmedOR31 and a common floral compound. Geraniol is also attractive to the cotton bollworm [8]. CmedOR32 belongs to the broad-spectrum OR receptors. It responded to a variety of volatile compounds, but the reaction current was weak (Figure 4g,h), which is a typical characteristic of broad-spectrum OR receptors. Ligand compounds of the three deorphaned OR genes are floral components attracting *C. medinalis*, conforming the attractiveness of floral blends to *C. medinalis* [34] at the molecular level. In the quantity of gene expression, we found that the expression level of CmedOR1 in *C. medinalis* was significantly higher in females than in males (Figure 5). The CmedOR1 ligand, phenylacetaldehyde, is a unique compound released by plants, and its level is closely related to the sugar level in nectar and pollen, while CmedOR1 plays an important role in locating flowers as nectar sources [42,53]. Phenylacetaldehyde is usually the major component of plant odor mixtures and has a wide range of attractiveness to lepidopteran species [39]. It may be a food indicator in the co-evolution between lepidopteran species and angiosperms [46]. We believe that OR1 is the odor receptor gene for *C. medinalis* females when searching for food. However, we did not find the ORs responded to methyl salicylate, another important component in the mixture attracting *C. medinalis* [3,34].

The migratory *C. medinalis* population in a particular location may be immigrants, local populations, emigrants, or a mix of these [54]. Adult *C. medinalis* with different physiological statuses during migration also have great differences in habitats/host selection, mating, and oviposition behaviors [55]. For migratory insects, their physiological and nutritional status and other biological and non-biological factors in the environment can impact their olfactory responses to sex pheromone and/or floral odor [34,56,57,58,59,60]. Our previous work revealed the relevancy of *C. medinalis* lured by the sex pheromone and floral odor under different physiological and nutritional status would affect the behavior of *C. medinalis* to mate or search for supplemental nutrition [34,56,57,58,59,60]. Many insect species, especially pollinators, have a close relationship in nutrition with their host flowers. However, insect olfactory responses to the scent of flowers depend on their developmental status for nutritional needs. Immigrant *C. medinalis* adults have a strong response to floral scents, probably due to their reduced body fat and the need for supplement nutrition in order to prolong their life span and optimize their egg production. The peak period of trapping from the floral lure was relatively short, with less than 10 days. Most trapped females were found to be mated or even oviposited [34]. Perhaps only a portion of adults in the population require supplementary nutrition. We found that the attractiveness of sex pheromone and floral odor to *C. medinalis* is related to its migration behavior. The olfactory response to the floral scent is also affected by its physiological status [34].

## 5. Conclusions

We identified the function of three olfactory receptor genes in *C. medinalis.* CmedOR31 was a specific receptor for geraniol, and CmedOR32 was a broad-spectrum OR receptor that responds to both foliar odor and floral scent. The ligands of CmedOR1 were linalool, phenethyl alcohol, 2-phenylethanol, and phenylacetaldehyde. CmedOR1 belonged to a unique conserved OR lineage that strongly responded to phenylacetaldehyde in the evolution of *Glossata* species, and the OR1 expression in *C. medinalis* antennae was significantly higher in females than in males. CmedOR1 was the odor receptor for *C. medinalis* adults to locate food sources.

## Figures and Tables

**Figure 1 insects-14-00930-f001:**
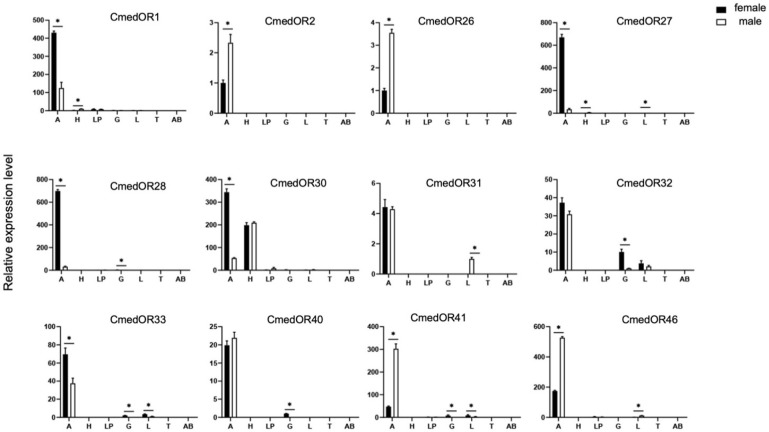
Relative expression levels of CmedOR genes in *C. medinalis* body parts. A: antennae; LP: labial palp; H: head; AB: abdomen; L: leg; G: genitals; T: thorax. Relative expression level is indicated as mean ± SE (*n* = 3). * indicates significant difference between male and female (*p* < 0.05).

**Figure 2 insects-14-00930-f002:**
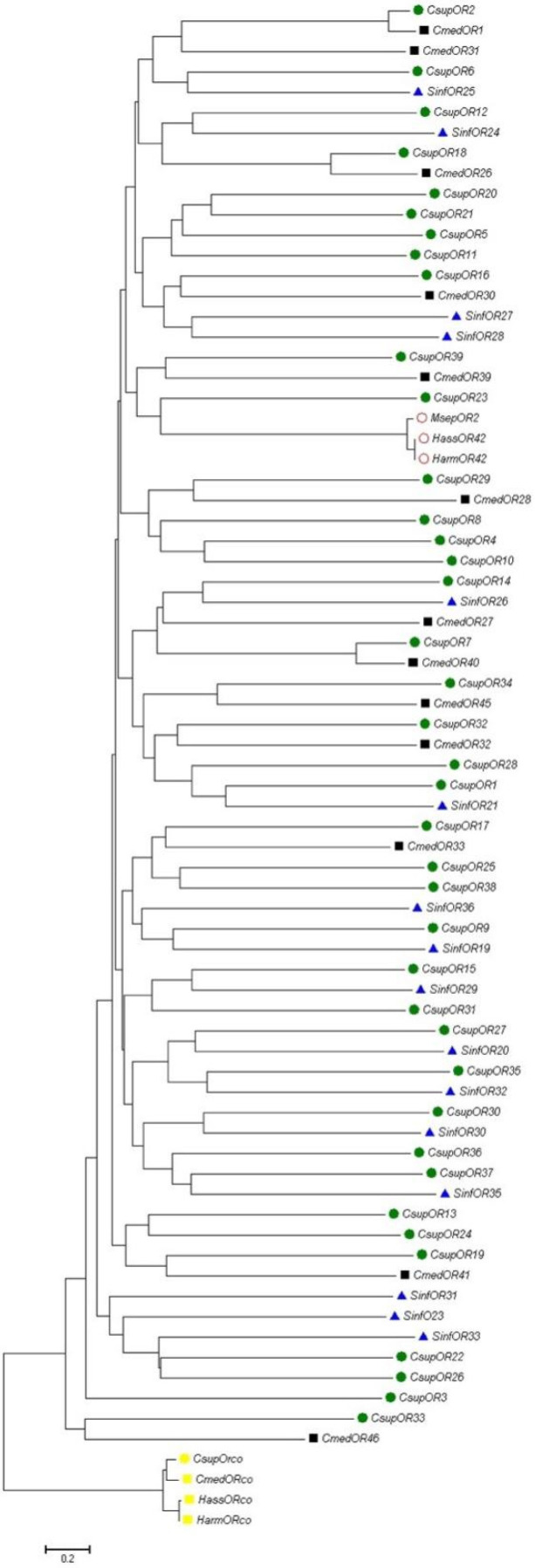
Phylogeny of lepidopteran ORs. The lepidopteran ORcos are defined as an out-group. The amino acid sequences were aligned with Clustal X 2.0, and the tree was constructed with MEGA7 using the neighbor-joining method. Values indicated at the nodes are bootstrap values based on 1000 replicates. Sinf: *Sesamia inferens*; Csup: *Chilo suppressalis*; Msep: *Mythimna separate*; Harm: *Helicoverpa armigera*; Hass: *Helicoverpa assulta*.

**Figure 3 insects-14-00930-f003:**
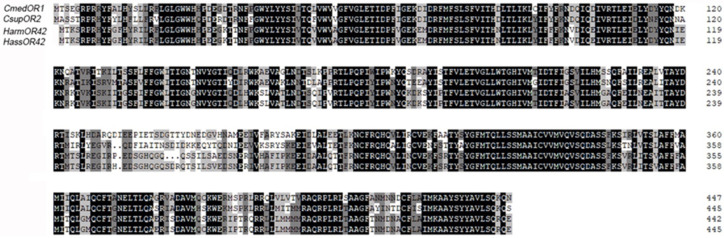
Multiple sequence alignment of CmedOR1. Multiple alignments and identity calculations were performed using DNAMAN 6.0 (Lynnon Biosoft, USA). Amino acid numbering is given on the right of the alignment.

**Figure 4 insects-14-00930-f004:**
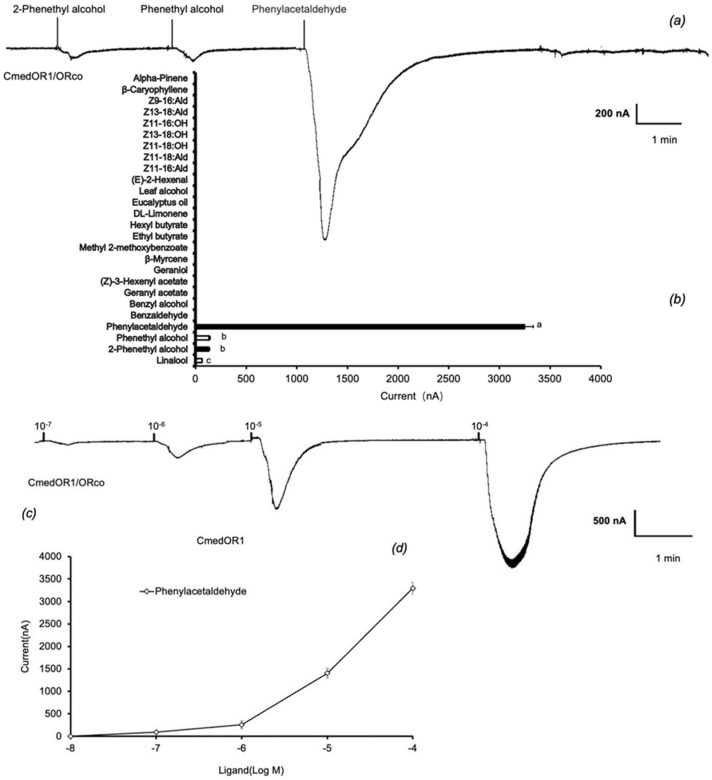
Functional characterization of CmedOR/CmedORco in Xenopus oocytes. (**a**) Inward currents of CmedOR1/ORco Xenopus oocytes in responses to chemical compounds at 10^–4^ M solution. (**b**) Response spectrum of CmedOR1/ORco Xenopus oocytes. (**c**) Responses of CmedOR1/ORco to phenylacetaldehyde at different dosages of each stimulus. (**d**) Inward currents of CmedOR1/ORco Xenopus oocytes in responses to phenylacetaldehyde at different dosages of each stimulus (**e**) Inward currents of CmedOR31/ORco Xenopus oocytes in responses to chemical compounds at 10^–4^ M solution. (**f**) Response spectrum of CmedOR31/ORco Xenopus oocytes. (**g**) Inward currents of CmedOR32/ORco Xenopus oocytes in responses to chemical compounds at 10^–4^ M solution. (**h**) Response spectrum of CmedOR32/ORco Xenopus oocytes.

**Figure 5 insects-14-00930-f005:**
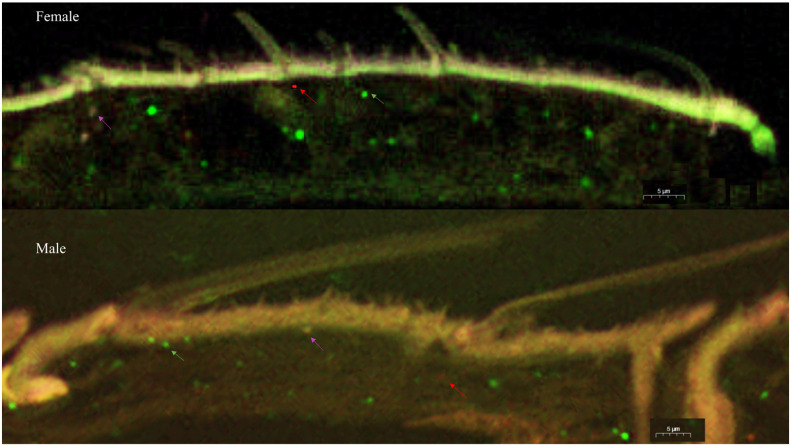
Fluorescence in situ hybridization of CmedOR genes. Three-color fluorescent in situ hybridizations of CmedORs showed different distributions in *C. medinalis* antennae. CmedOR1 (green signal), CmedOR31 (red signal), and CmedOR32 (pink signal).

**Table 1 insects-14-00930-t001:** Name, source, and other information about plant odor compounds used in the test.

Volatile Compounds	CAS	Purity	Manufacturer
Linalool	78-70-6	98.00%	Aladdin (Shanghai, China)
Z3-C6:OH	928-96-1	98.00%	Aladdin (Shanghai, China)
Z3-C6:Ald	6789-80-6	50% intriacetin	Sigma-Aldrich (Shanghai, China)
Phenylacetaldehyde	122-78-1	95.00%	Aladdin (Shanghai, China)
Benzaldehyde	100-52-7	99.00%	Aladdin (Shanghai, China)
β-Caryophyllene	87-44-5	98.00%	Sigma-Aldrich (Shanghai, China)
β-Myrcene	123-35-3	90.00%	Aladdin (Shanghai, China)
alpha-Pinene	80-56-8	98.00%	Aladdin (Shanghai, China)
beta-Pinene	2437-95-8	98.00%	Macklin (Shanghai, China)
Methyl salicylate	119-36-8	99.00%	Aladdin (Shanghai, China)
Geraniol	106-24-1	98.00%	Aladdin (Shanghai, China)
Indole	120-72-9	99.00%	Aladdin (Shanghai, China)
Geranyl acetate	105-87-3	95.00%	Aladdin (Shanghai, China)
Benzyl alcohol	100-51-6	99.00%	Aladdin (Shanghai, China)
Phenethyl alcohol	60-12-8	99.50%	Aladdin (Shanghai, China)
E2-C6:Ald	6728-26-3	98.00%	Aladdin (Shanghai, China)
Leaf alcohol	928-96-1	98.00%	Macklin (Shanghai, China)
cis-3-Hexenyl acetate	3681-71-8	98.00%	Sigma-Aldrich (Shanghai, China)
Methyl 2-methoxybenzoate	606-45-1	98.00%	Aladdin (Shanghai, China)
Isolongifolene	1135-66-6	98.00%	Sigma-Aldrich (Shanghai, China)
Hexyl butyrate	2639-63-6	98.00%	Aladdin (Shanghai, China)
DL-Limonene	138-86-3	95.00%	Aladdin (Shanghai, China)
Methyl phenylacetate	101-41-7	99.00%	Aladdin (Shanghai, China)
Eucalyptus oil	8000-48-4	80.00%	Macklin (Shanghai, China)
Ethyl butyrate	105-54-4	99.50%	Aladdin (Shanghai, China)
β-ionone	8013-90-9	90.00%	Sigma-Aldrich (Shanghai, China)
Z11-18:Ald	4273-95-4	97.00%	Newcon, Inc. (Ningbo, China)
Z11-18:OH	62803-19-4	97.00%	Newcon, Inc. (Ningbo, China)
Z13-18:Ald	58594-45-9	97.00%	Newcon, Inc. (Ningbo, China)
Z13-18:OH	69820-27-5	97.00%	Newcon, Inc. (Ningbo, China)
Z11-16:Ald	53939-28-9	98.00%	Newcon, Inc. (Ningbo, China)
Z9-16:Ald	56219-04-6	98.00%	Newcon, Inc. (Ningbo, China)

## Data Availability

The data presented in this study are available on request from the corresponding author.

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
