# Peer review of "Functional Identification of Olfactory Receptors of Cnaphalocrocis medinalis (Lepidoptera: Crambidae) for Plant Odor"

_insects, 2023, doi:10.3390/insects14120930_

Round 1
Reviewer 1 Report
Comments and Suggestions for Authors
The rice leaf roller Cnaphalocrocis medinalis is a migratory insect feeding the rice leaf. Therefore, management of this pest is essential to yield of the rice. Many strategies can be employed in the control of pests. Plant odor-trapping has been a tool for monitoring the population of Cnaphalocrocis medinalis playing an important role in the pest management system. It is still unknown in the mechanism underlying how C. medinalis recognizes the plant odor. This study investigated 12 odor receptors from this insect and identified their expression level and found their functions in responding to different odors. Their results provide further insights into the mechanism of how odors attract insect through odor receptors. Although the manuscript was written well and results were presented clearly, I still have some concerns, hoping with help in strengthening the manuscript. Below are my comments.
Major comments.
1. I agree that authors showed evidence of these 12 odor receptor genes in sensing the odors; why did authors focus on these 12 receptors rather than other receptor genes ? How were these genes selected? Did authors have the screen of gene candidates ahead? Authors should give some explanation on how they focused on these 12 gene receptors.
2. In the functional responses to odor tests, authors expressed these odor receptor genes in Xenopus oocyteexpression system to electrophysiologically record the current responses. Although current responses are clear and obvious before and after the addition of odors, the control only with Xenopus oocyte cells with empty expression of these receptors is missing. Authors need to show the data of how Xenopus oocyte cells respond to these odors without any receptor expressions, which will be more convincing.
3. In Fig.5 about the situ hybridization, authors only showed the fluorescence images. If possible, it will be more helpful when authors can do some quantifications such as through region of interest. Additionally, pink signal is very difficult to identify, so it will be helpful if some hints like arrows can be added.
Minor comments.
1. About the electrophysiology of methods, what electrodes were used? What is the electrode resistance after filling with the internal solution? What is the sampling frequency? What is the access resistance after the whole-cell configuration? These parameters should be indicated for electrophysiology.
2. Lines 207 ,209, 212, 215, “Fig” should be “Figs” in brackets because not one figure is included. Authors can go through the manuscript to find similar cases.
Author Response
Major comments.
1.I agree that authors showed evidence of these 12 odor receptor genes in sensing the odors; why did authors focus on these 12 receptors rather than other receptor genes ? How were these genes selected? Did authors have the screen of gene candidates ahead? Authors should give some explanation on how they focused on these 12 gene receptors.
Response: “We used three criteria to filter OR genes in this study: high expression levels, long fragments and homology analysis of amino acid sequences with other insects ORs whose function have been identified. However, the general odor receptors for detecting plant volatiles have low homology, which are different from sex pheromone receptor genes in conservation. Except for CmedOR1, which we found, and is a unique conserved general odor receptor gene, the rest of the ORs in C. medinalis moths does not have much common in the homology of genes, so in order to study the odor recognition mechanism in C. medinalis to plant volatiles, it is necessary to test different OR genes. Overall, we selected these 12 OR genes in the present study” has been added to the Discussion section.
2.In the functional responses to odor tests, authors expressed these odor receptor genes in Xenopus oocyteexpression system to electrophysiologically record the current responses. Although current responses are clear and obvious before and after the addition of odors, the control only with Xenopus oocyte cells with empty expression of these receptors is missing. Authors need to show the data of how Xenopus oocyte cells respond to these odors without any receptor expressions, which will be more convincing.
Response: “Xenopus laevis endogenous receptors and channels require the existence of follicular membrane. In our experiments, we performed pre-treatment on Xenopus laevis oocytes such as collagenolysis to eliminate the follicular membrane, so when we were conducting compound testing, all current responses of the blank group were 0.” has been added to the “2.5. Electrophysiological recording” method section. .
3.In Fig.5 about the situ hybridization, authors only showed the fluorescence images. If possible, it will be more helpful when authors can do some quantifications such as through region of interest. Additionally, pink signal is very difficult to identify, so it will be helpful if some hints like arrows can be added.
Response: Fig. 5 has been replaced with a new one with better image and arrows.
Minor comments.
1.About the electrophysiology of methods, what electrodes were used? What is the electrode resistance after filling with the internal solution? What is the sampling frequency? What is the access resistance after the whole-cell configuration? These parameters should be indicated for electrophysiology.
Response: “Two electrodes are both glass electrodes with the tip diameter 3 to 7 μm. The glass electrode was filled with 3 mol/L KCl solution, and the electrode resistance was 0.1~1 M glass electrode was filled with 3 mol/L KCl solution, and the electrode resistance was 0.1tance after filling with the inter” has been added to the “2.5. Electrophysiological recording” section.
2.Lines 207 ,209, 212, 215, “Fig” should be “Figs” in brackets because not one figure is included. Authors can go through the manuscript to find similar cases.
Response: Lines 207, 209, 212 and 215, “Fig” has been changed to “Figs”.
Reviewer 2 Report
Comments and Suggestions for Authors
The reviewer has read with much interest the manuscript entitled as ‘Functional identification of olfactory receptors of Cnaphalocrocis medinalis (Lepidoptera: Crambidae) to plant odor’ submitted by Jianjun Cheng , Jiawei Gui , Xiaoming Yao , Hong Zhao , Yujie Zhou , Yongjun Du to INSECTS. The authors studied the antennal olfactory receptors of Cnaphalocrocis medinalis in molecular level, identified genes of 12 ORs and particularly examined 3 ORs (CmedOR1, CmedOR31 and CmedOR32) which were expressed stronger than other 9 ORs. Further, the authors examined phylogenetic relations of the ORs and found that it belongs to a uniquely conserved OR pedigree in the evolution of Glossata species and that the ORs of this pedigree strongly responded to phenylacetaldehyde.
The manuscript is fundamentally well written and the conclusions seem appropriate. However, some problems are present in the manuscript as follows.
Lines 13-14: The reviewer thinks that it is not always necessary to know the molecular mechanism for plant odor-trapping and for the time being enough to know only the response spectral features of olfactory receptors. Thus, please describe more adequate reasons for the planning and conduct of this study.
Line 41: Functional identification -> Change to Roman style.
Line 63: transduce to --> are transmitted to
Line 65: olfactory receptors -->organs -->sensilla
Line 72: lepidopteran --> lepidoptera
Line 82: C. medinalis adults were captured --> Cnaphalocrocis medinalis adults used in this study were captured
Lines 83-84: The adult antennae, head, thorax, abdomen, leg, labipalpus, and genitalia --> The adult antennae, head, thorax, abdomen, legs, labipalpi, and genitalia
Line 93: Generation (Roche, USA), --> Generation (Roche, USA)
Lines 179-187: This part is very interesting!
Line 208: (10−7 M) --> (10-7M)
Lines 239-270: CmedOR1 was shown in the long trichoid sensilla in males but in the short trichoid sensilla in females. ---> The authors referred to no papers on morphology of antennal sensilla. The authors should refer to papers which describe structure of antennae and antennal sensilla of this species and closed related species. The reviewer does not know that the same receptor protein is expressed in the receptor neurons housed in the morphologically different type of sensilla in adult insects. Do the authors know such phenomena in other insects?
Figure 5: Although the two micrographs are shown, their magnification ratios are different each other. Please replace these micrographs to the same magnified ones.
Line 253: 10−4 M --> 10−4 M
Lines 275-278: Therefore, understanding the molecular mechanism of olfactory recognition in C. medinalis to floral odors, especially the function of olfactory receptors, would be helpful to improve the pest monitoring method. ---> Please explain this idea in more detail.
Reference 12., 22., 34., 59. ---> Please check format.
Lastly, the olfactory receptors which the authors identified in molecular level respond to odors which are not so specific. Do the authors think which is the mechanism to attract
C. medinalis to the host plant among the host plant specific odors and the specific combination of host plant odors?
Author Response
The manuscript is fundamentally well written and the conclusions seem appropriate. However, some problems are present in the manuscript as follows.Lines 13-14: The reviewer thinks that it is not always necessary to know the molecular mechanism for plant odor-trapping and for the time being enough to know only the response spectral features of olfactory receptors. Thus, please describe more adequate reasons for the planning and conduct of this study.
Response: In P12, “Floral odor attracts both female and male moths while the sex pheromone can only attract males. However, our previous research have demonstrated that for C. medinalis female moths in the immigratory and the local generations, mostly trapped by the floral odor were those mated and egg-laid, whereas the females releasing sex pheromone to attract males were those in their calling period [34]. Therefore, the physiological and migratory status of adult C. medinalis moths in the field could be differentiated by two types of attractants, namely the floral odor and the sex pheromone.” has been add to the Discussion section.
Line 41: Functional identification -> Change to Roman style.
Response: Line 41 has been changed to Roman style.
Line 63: transduce to --> are transmitted to
Response: “transduce” has been changed to “transmitted”.
Line 65: olfactory receptors -->organs -->sensilla
Response: It has been changed to “sensilla”.
Line 72: lepidopteran --> lepidoptera
Response: “lepidopteran” has been changedto “lepidiptera”.
Line 82: C. medinalis adults were captured --> C naphalocrocis medinalis adults used in this study were captured
Response: “C. medinalis adults were captured” has been changed to “C naphalocrocis medinalis adults used in this study were captured”.
Lines 83-84: The adult antennae, head, thorax, abdomen, leg, labipalpus, and genitalia --> The adult antennae, head, thorax, abdomen, legs, labipalpi, and genitalia
Response: “The adult antennae, head, thorax, abdomen, leg, labipalpus, and genitalia” has been changed to “The adult antennae, head, thorax, abdomen, legs, labipalpi, and genitalia”.
Line 93: Generation (Roche, USA), --> Generation (Roche, USA)
Response: “Generation (Roche, USA)” has been changed to “Generation (Roche, USA)”.
Lines 179-187: This part is very interesting!
Response: Thank you very much!
Line 208: (10−7 M) --> (10-7M)
Response: “10−7 M” has been changed to “10-7M”.
Lines 239-270: CmedOR1 was shown in the long trichoid sensilla in males but in the short trichoid sensilla in females. ---> The authors referred to no papers on morphology of antennal sensilla. The authors should refer to papers which describe structure of antennae and antennal sensilla of this species and closed related species. The reviewer does not know that the same receptor protein is expressed in the receptor neurons housed in the morphologically different type of sensilla in adult insects. Do the authors know such phenomena in other insects?
Response: “Sun, X.; Wang, M. Q.; Zhang, G., Ultrastructural observations on antennal sensilla of Cnaphalocrocis medinalis (Lepidoptera: Pyralidae). Microscopy Research & Technique 2011, 74, 113-121.” has been added to the References section. In addition, “According to previous study on the sensilla of C. medinalis [36], CmedOR1 was shown in the trichoid sensilla both in males and females.” has been added in P9.According to the literature, the sensilla of C. medinalis moths have three different sensilla trichodea. However, since fluorescence in situ hybridization experiments need sectioning and washing, which destroyed the original shape of the sensilla. Therefore, it is difficult to compare the sensilla under CLSM (confocal laser scanning microscope) and under the SEM(scanning electron microscope). In order to express it more accurately, the sensilla are not distinguished here, and they are all called sensilla trichodea.
Figure 5: Although the two micrographs are shown, their magnification ratios are different each other. Please replace these micrographs to the same magnified ones.
Response: Since the two micrographs were not in the same level in the tissue, it would be hard to have the maginification at the same level.
Line 253: 10−4 M --> 10−4 M
Response: “10−4 M” has been changed to “10−4 M”.
Lines 275-278: Therefore, understanding the molecular mechanism of olfactory recognition in C. medinalis to floral odors, especially the function of olfactory receptors, would be helpful to improve the pest monitoring method. ---> Please explain this idea in more detail.
Response: In P12, “Floral odor attracts both female and male moths while the sex pheromone can only attract males. However, our previous research have demonstrated that for C. medinalis female moths in the immigratory and the local generations, mostly trapped by the floral odor were those mated and egg-laid, whereas the females releasing sex pheromone to attract males were those in their calling period [34]. Therefore, the physiological and migratory status of adult C. medinalis moths in the field could be differentiated by two types of attractants, namely the floral odor and the sex pheromone.” has been added.
Reference 12., 22., 34., 59. ---> Please check format.
Response:
Reference 12 has been changed to “Heath, R. R.; Landolt, P. J.; Dueben, B.; Lenczewski, B., Identification of Floral Compounds of Night-Blooming Jessamine Attractive to Cabbage-Looper Moths. Environ. Entomol. 1992, 21, 854-859.”
Reference 22 has been changed to “Leal, W.S. Odorant Reception in Insects: Roles of Receptors, Binding Proteins, and Degrading Enzymes. Annu. Rev. Entomol.2013, 58, 373-391.”
Reference 34 has been changed to “Guo, Q.S.; Zhuo, F.Y.; Zhu, J.Q.; Lin, Y.F.; Zhang, Z.B.; Huang, D.C.; Zhang, S.L.; Du Y.J., Responses of the rice leaffolder, Cnaphalocrocis medinalis (Lepidoptera Crambidae) to sex pheromone and floral odor in olfactory behavior, and their application in its population monitoring. Acta Entomologica Sinica 2022, 65, 289-303.”
Reference 60 has been changed to “Gadenne, C.; Renou, M.; Sreng, L., Hormonal control of pheromone responsiveness in the male black cutworm. Agrotis ipsilon. Experientia.1993, 49, 721-724.”
Lastly, the olfactory receptors which the authors identified in molecular level respond to odors which are not so specific. Do the authors think which is the mechanism to attract C. medinalis to the host plant among the host plant specific odors and the specific combination of host plant odors?
Response: The mechanism may be very complicated: e.g. 1) In addition to the odorants and receptors identified in the present study, there may be other receptors that have not been identified, and 2) a mixture of multiple compounds in a certain ratio may play a role.
Round 2
Reviewer 1 Report
Comments and Suggestions for Authors
Authors resolved my concerns reasonably. This manuscript now is more convincing. I advise this manuscript can be accepted for publication.